

# Prediction of individual mortality risk among patients with chronic obstructive pulmonary disease: a convenient, online, individualized, predictive mortality risk tool based on a retrospective cohort study

Shubiao Lu[1,*], Yuwen Zhou[2,*], Xuejuan Huang[3], Jinsong Lin[1], Yingyu Wu[1] and Zhiqiao Zhang[1]

[1] Department of Internal Medicine, The Affiliated Chencun Hospital of Shunde Hospital, Southern Medical University, Shunde, Guangdong, China
[2] Emergency Department, The Affiliated Chencun Hospital of Shunde Hospital, Southern Medical University, Shunde, Guangdong, China
[3] Obstetrics and Gynecology Department, The Affiliated Chencun Hospital of Shunde Hospital, Southern Medical University, Shunde, Guangdong, China
[*] These authors contributed equally to this work.

Corresponding authors
Jinsong Lin, xuebeibei6474@126.com
Zhiqiao Zhang, sdgrxjbk@163.com

## ABSTRACT

**Background.** Chronic obstructive pulmonary disease (COPD) is a serious condition with a poor prognosis. No clinical study has reported an individual-level mortality risk curve for patients with COPD. As such, the present study aimed to construct a prognostic model for predicting individual mortality risk among patients with COPD, and to provide an online predictive tool to more easily predict individual mortality risk in this patient population.

**Patients and methods.** The current study retrospectively included data from 1,255 patients with COPD. Random survival forest plots and Cox proportional hazards regression were used to screen for independent risk factors in patients with COPD. A prognostic model for predicting mortality risk was constructed using eight risk factors.

**Results.** Cox proportional hazards regression analysis identified eight independent risk factors among COPD patients: B-type natriuretic peptide (hazard ratio [HR] 1.248 [95% confidence interval (CI) 1.155–1.348]); albumin (HR 0.952 [95% CI 0.931–0.974]); age (HR 1.033 [95% CI 1.022–1.044]); globulin (HR 1.057 [95% CI 1.038–1.077]); smoking years (HR 1.011 [95% CI 1.006–1.015]); partial pressure of arterial carbon dioxide (HR 1.012 [95% CI 1.007–1.017]); granulocyte ratio (HR 1.018 [95% CI 1.010–1.026]); and blood urea nitrogen (HR 1.041 [95% CI 1.017–1.066]). A prognostic model for predicting risk for death was constructed using these eight risk factors. The areas under the time-dependent receiver operating characteristic curves for 1, 3, and 5 years were 0.784, 0.801, and 0.806 in the model cohort, respectively. Furthermore, an online predictive tool, the "Survival Curve Prediction System for COPD patients", was developed, providing an individual mortality risk predictive curve, and predicted mortality rate and 95% CI at a specific time.

**Conclusion**. The current study constructed a prognostic model for predicting an individual mortality risk curve for COPD patients after discharge and provides a convenient online predictive tool for this patient population. This predictive tool may provide valuable prognostic information for clinical treatment decision making during hospitalization and health management after discharge (https://zhangzhiqiao15. shinyapps.io/Smart_survival_predictive_system_for_COPD/).

## INTRODUCTION

Chronic obstructive pulmonary disease (COPD) is a common chronic airway condition that seriously affects the quality of life of affected individuals. It has been estimated that COPD is the third most common cause of death globally (*Lozano et al., 2012*). The prevalence of COPD in adults ≥ 20 years of age is approximately 8.6%, whereas the prevalence among those >40 years of age is as high as 13.7% (*Wang et al., 2018*). The prevalence rate of stage ≥ II COPD can reach 10.1% in the general population (*Buist et al., 2007*). The three-year mortality rate of COPD patients has been reported to be 10.0%–36.9% according to the Global Initiative for Obstructive Lung Disease (*i.e.*, "GOLD") 2017 classification criteria (*Gedebjerg et al., 2018*). More than 3.2 million individuals die from COPD annually (*anonymous, 2017*). Therefore, COPD is a serious public health challenge that requires urgent attention from government departments and medical institutions.

Several prognostic models have been developed to predict prognosis among COPD patients, including B-AE-D-C (*Boeck et al., 2016*), extended ADO (*Puhan et al., 2012*), ADO (*Puhan et al., 2009*), updated BODE (*Puhan et al., 2009*), PSI (*Hu et al., 2015*), and CURB65 (*Chang et al., 2011*). These previous models divide patients into high- and low-risk groups and evaluate mortality risk in different groups. To the best of our knowledge, these models are not able to describe individualized survival curves for specific patients. One Japanese research team established a prognostic model for predicting mortality among patients who experience acute exacerbation of COPD during hospitalization (*Sakamoto et al., 2017*). Although this study could predict the risk for death during hospitalization based on individual patient information, it did not further provide a predictive mortality curve for individual patients during the follow-up period after discharge.

With the development of "big data" analytics and data mining algorithms, precision medicine has witnessed significant advances in several research fields. Several precision medicine studies have been able to predict individual mortality curves for specific patients based on clinical information and have provided convenient predictive web tools for patients (*Zhang et al., 2021*; *He et al., 2021*; *Lin et al., 2021*). This convenient, predictive web tool could help patients better evaluate the risk for death and reasonably facilitate individual treatment decisions.

The current study aimed to construct a prognostic model for predicting mortality risk in individual COPD patients based on baseline characteristics. Furthermore, we plan to develop and maintain an online tool to provide an individualized, predictive mortality risk curve for patients with COPD.

## METHOD

### Patients

COPD patients hospitalized in the Department of Respiratory Medicine of Shunde Hospital, Southern Medical University (Foshan City, Guangzhou Province, China) and the Department of Internal Medicine of The Affiliated Chencun Hospital of Shunde Hospital, Southern Medical University, between September 2009 and December 2019, were included. All patients underwent pulmonary function examination after inhaling a tracheal dilator before enrollment and were diagnosed according to a forced expiratory volume in 1 s ($FEV_1$)/forced vital capacity <70%, which fulfilled the diagnostic criteria for chronic obstructive pulmonary emphysema ($n = 1309$). The deadline for follow-up of enrolled patients was May 1, 2020. Patients with missing survival time data were excluded from the survival analysis ($n = 54$). This study was reviewed and approved by the Ethics Committee of The Affiliated Chencun Hospital of Shunde Hospital, Southern Medical University (ID: 202202001). Due to the retrospective nature of the study and the use of anonymized data, requirements for informed consent were waived by the Ethics Committee of The Affiliated Chencun Hospital of Shunde Hospital, Southern Medical University (ID: 202202001). The current study was conducted in accordance with the Declaration of Helsinki, relevant guidelines, and local regulations.

### Information collection

The following information was collected and recorded for survival analysis: general information, including age, sex, body mass index (BMI), smoking history, and smoking time; clinical/biochemical results within 24 h after admission, including body temperature, systolic blood pressure, diastolic blood pressure, heart rate, respiratory rate, respiratory index, partial pressure of arterial oxygen ($Pa\,O_2$), pH, oxygenation index, partial pressure of arterial carbon dioxide ($Pa\,CO_2$), sodium, potassium, calcium, blood urea nitrogen (BUN), creatinine, serum albumin (ALB), serum globulin (GLB), C-reactive protein, and B-type natriuretic peptide (BNP) levels, platelets, white blood cell count, granulocyte ratio (GR), blood glucose, and state of consciousness. Original BNP values were converted into an ordered hierarchical variable according to the expert consensus on BNP clinical application recommendations published by the American College of Cardiology (*Silver et al., 2004*), as follows: no heart failure (BNP <80 ng/L); grade I heart failure (BNP 95–221 ng/L); grade II heart failure (BNP 221–459 ng/L); grade III heart failure (BNP 459–1006 ng/L); and grade IV heart failure (BNP >1006 ng/L); and effective follow-up, the end date of follow-up in this study was May 1, 2020. The survival time of deceased patients was calculated by subtracting the date of discharge from the date of death. The survival time of surviving patients was calculated by subtracting the discharge date from May 1, 2020.

## Model building

The current study constructed a prognostic predictive model for COPD patients using a Cox proportional hazards regression algorithm. The Cox proportional hazards regression model is a semi-parametric regression model, which uses survival outcome and survival time as dependent variables, and can analyze the independent effects of multiple factors on survival outcome and survival time at the same time (*Fisher & Lin, 1999*; *Moolgavkar et al., 2018*). As a semi-parametric regression model, the Cox proportional hazards regression model could be used to analyze data with censored survival time. The Cox proportional hazards regression model has been widely used to construct prognostic models for various diseases (*Han et al., 2021*; *Royston & Altman, 2013*; *Luo et al., 2017*).

## Statistical analysis

Statistical analysis in the current study was performed using R version 3.6.1 (*R Core Team, 2019*). Continuous variables were compared between the two groups (*i.e.,* model *versus* validation) using the $t$-test for data that were normally distributed, while the Mann–Whitney U test was used for data that were not normally distributed. The chi-squared test (default method for contingency table analysis) or Fisher's exact probability method (in case any grid was found to be <1) was used to compare categorical variables between the two groups. The random survival forest method is used to identify valuable risk factors for prognosis (*Hsich et al., 2011*). Differences with $P < 0.05$ were considered to be statistically significant.

# RESULTS

## Baseline characteristics

A total of 1,255 patients were ultimately included in the current analysis and were divided into a model cohort ($n = 627$) and a validation cohort ($n = 628$) using a random sampling method. The mortality rate was 78.3% (491/627) in the model cohort and 76.6% (481/628) in the validation cohort ($P = 0.509$). A comparison of baseline characteristics between the model and validation cohorts is summarized in Table 1.

## Variable selection

The relative importance of various independent variables was explored using the random survival forest algorithm (Fig. 1). The top three important variables included BNP, age, and *Pa* CO$_2$. Subsequently, BNP (hazard ratio [HR] 1.248 [95% CI [1.155–1.348]]), albumin (HR 0.952 [95% CI [0.931–0.974]]), age (HR 1.033 (95% CI [1.022–1.044])), GLB (HR 1.057 (95% CI [1.038–1.077]), smoking years (HR 1.011 (95% CI [1.006–1.015]), *Pa* CO$_2$ (HR 1.012 (95% CI [1.007–1.017])), granulocyte ratio (HR 1.018 (95% CI [1.010–1.026])), and BUN (HR 1.041 (95% CI [1.017–1.066])) were identified to be independent risk factors for the prognosis of COPD patients according to multivariate Cox proportional hazards regression analysis (Fig. 2, Table 2).

**Table 1** Baseline characteristics between model cohort and validation cohort.

| Parameter | Stratification | Total | Model cohort | Validation cohort | Test_value | P _Value |
|---|---|---|---|---|---|---|
| Survival_status [n(%)] | 0 | 283(22.5) | 136(10.8) | 147(11.7) | 0.44 | 0.509 |
| | 1 | 972(77.5) | 491(39.1) | 481(38.3) | | |
| Gender [n(%)] | 0 | 373(29.7) | 161(12.8) | 212(16.9) | 9.42 | 0.002 |
| | 1 | 882(70.3) | 466(37.1) | 416(33.2) | | |
| Pathogens [n(%)] | 0 | 1218(97.1) | 605(48.2) | 613(48.8) | 1.01 | 0.314 |
| | 1 | 37(2.9) | 22(1.8) | 15(1.2) | | |
| Consciousness [n(%)] | 0 | 1180(94.0) | 581(46.3) | 599(47.7) | 8.56 | 0.073 |
| | 1 | 53(4.2) | 32(2.6) | 21(1.7) | | |
| | 2 | 16(1.3) | 8(0.6) | 8(0.6) | | |
| | 3 | 5(4.0) | 5(0.4) | 0(0) | | |
| | 4 | 1(0.1) | 1(0.1) | 0(0) | | |
| Center [n(%)] | 1 | 99(7.9) | 48(3.8) | 51(4.1) | 0.04 | 0.841 |
| Smoking_year | | 30.0(0.0,40.0) | 30.0(0.0,40.0) | 30.0(0.0,45.0) | 185598 | 0.073 |
| Smoking_index | | 500.0(0.0,1000.0) | 425.0(0.0,1000.0) | 600.0(0.0,1025.0) | 183529.5 | 0.034 |
| Age (Year) | | 74.0(67.0,79.0) | 74.0(67.0,79.0) | 74.0(66.0,79.0) | 198290 | 0.826 |
| Weight (Meter) | | 51.0(45.0,55.0) | 51.0(45.0,55.0) | 51.0(45.0,56.0) | 196422.5 | 0.943 |
| Height (Meter) | | 1.62(1.58,1.65) | 1.62(1.57,1.65) | 1.62(1.58,1.65) | 190905 | 0.351 |
| BMI | | 19.2(17.7,21.1) | 19.1(17.7,21.1) | 19.2(17.7,21.0) | 198432.5 | 0.809 |
| Temperature (°C) | | 36.9(36.8,37.2) | 37.0(36.8,37.2) | 36.9(36.7,37.2) | 210293 | 0.035 |
| HR | | 100.0(90.0,112.0) | 100.0(90.0,112.0) | 100.0(90.0,110.0) | 194186.5 | 0.675 |
| RR | | 22.0(22.0,26.0) | 22.0(22.0,24.0) | 22.0(22.0,26.0) | 188351.5 | 0.173 |
| DBP (mmHg) | | 132.0(120.0,143.0) | 132.0(121.0,143.0) | 131.0(118.0,143.0) | 204455 | 0.238 |
| SBP (mmHg) | | 79.6 ± 12.3 | 79.9 ± 11.8 | 79.3 ± 12.8 | 0.894 | 0.372 |
| WBC (109/L) | | 8.6(6.4,11.9) | 8.6(6.4,11.9) | 8.5(6.3,11.9) | 196569 | 0.962 |
| GR (%) | | 75.1(65.9,83.0) | 75.1(66.1,83.0) | 75.3(65.3,83.2) | 198623.5 | 0.786 |
| HB (g/L) | | 132.0(118.0,145.0) | 131.5(116.0,144.0) | 132.0(119.0,145.0) | 191450 | 0.398 |
| PLT (109/L) | | 229.0(172.0,293.0) | 227.0(174.0,292.0) | 229.0(170.5,294.0) | 196630 | 0.969 |
| K (mmol/L) | | 4.0(3.7,4.4) | 4.0(3.7,4.4) | 4.0(3.7,4.4) | 193360 | 0.583 |
| Na (mmol/L) | | 141.0(137.0,144.0) | 141.0(137.0,145.0) | 141.0(137.0,144.0) | 202970.5 | 0.342 |
| Ca (mmol/L) | | 2.3(2.2,2.4) | 2.3(2.2,2.4) | 2.3(2.2,2.4) | 198154 | 0.842 |
| Glu (mmol/L) | | 6.4(5.4,8.3) | 6.4(5.4,8.4) | 6.4(5.4,8.1) | 200859.5 | 0.535 |
| Bun (mmol/L) | | 5.3(4.1,7.1) | 5.2(4.1,6.9) | 5.4(4.1,7.2) | 189045 | 0.222 |
| Cr (mmol/L) | | 80.8(66.3,99.6) | 80.5(65.8,98.3) | 81.0(66.7,101.0) | 192127 | 0.459 |
| PH | | 7.359(7.317,7.397) | 7.358(7.318,7.393) | 7.36(7.317,7.399) | 193648.5 | 0.615 |
| PaCO$_2$ (mmHg) | | 47.8(41.7,59.9) | 47.7(41.3,59.2) | 47.9(41.9,60.1) | 190215 | 0.299 |
| PO2 (mmHg) | | 81.9(64.9,103.2) | 82.3(64.7,104.9) | 81.1(65.2,101.5) | 202484.5 | 0.383 |
| Oxygen_index | | 287.1(232.8,342.8) | 285.6(232.8,339.1) | 289.7(232.9,347.0) | 195917.5 | 0.881 |
| AST (U/L) | | 25.0(20.0,35.0) | 25.0(20.0,35.0) | 25.0(20.0,34.0) | 197225 | 0.957 |
| ALT (U/L) | | 15.0(10.0,25.0) | 15.0(10.8,26.0) | 15.0(10.0,25.0) | 204637.5 | 0.226 |
| ALB (g/L) | | 38.8(36.1,41.5) | 38.8(36.1,41.5) | 38.9(36.0,41.6) | 194930.5 | 0.762 |
| GLB (g/L) | | 23.7(20.5,26.7) | 23.7(20.6,26.5) | 23.7(20.5,27.0) | 194228.5 | 0.68 |
| CRP (μg/L) | | 24.4(6.6,67.0) | 25.0(6.7,71.6) | 22.4(6.4,69.0) | 201399 | 0.481 |
| Survival_month (month) | | 38.5(11.1,92.0) | 41.1(12.9,94.4) | 34.4(10.3,86.7) | 209713.5 | 0.046 |
| BNP (ng/L) | | 65.5(28.6,231.1) | 62.45(29.0,221.2) | 70.2(28.3,236.2) | 193919.5 | 0.645 |

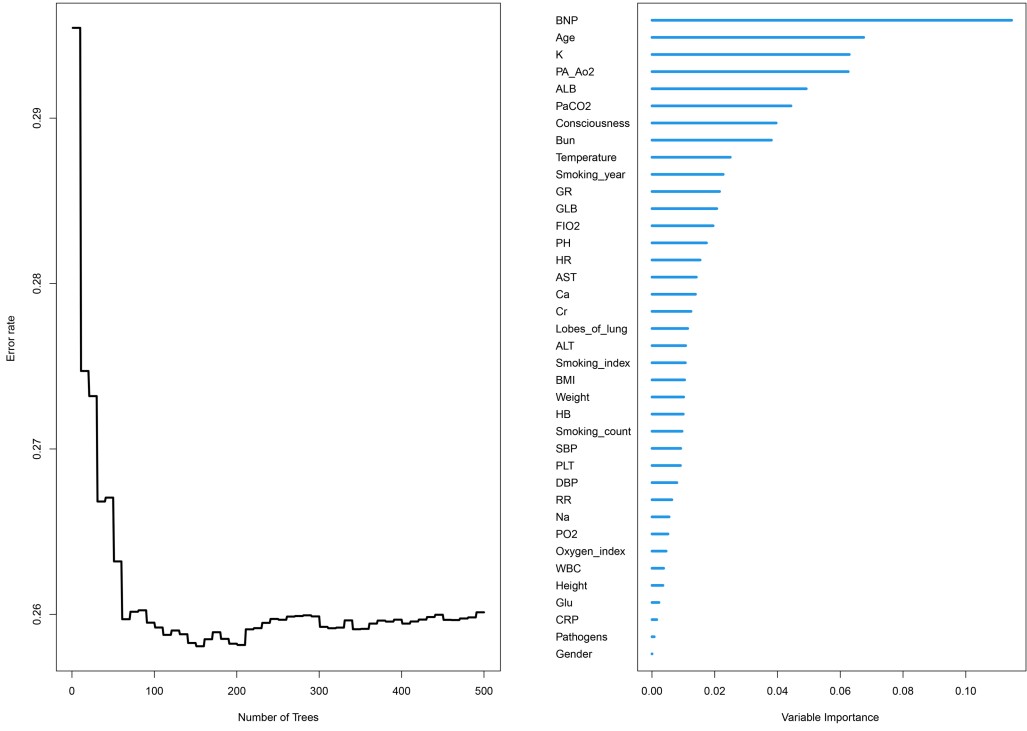

**Figure 1** Relative importance and error rate curve of variables by using the random survival forest algorithm.

## Prognostic predictive model

According to the results of multivariate Cox proportional hazards regression analysis, a prognostic predictive model for COPD patients was established using the following equation:

Prognostic score = 0.221*BNP [ng/L] −0.049*ALB [g/L] + 0.033*Age [years] + 0.056*GLB [g/L] + 0.01*smoking years [years] + 0.012*$Pa$CO$_2$ [mmHg] + 0.018*GR [%] + 0.04*BUN [mmol/L].

A prognostic predictive nomogram chart is presented in Fig. 3 according to the results of multivariate Cox proportional hazards regression analysis.

## Performance in the model cohort

The area under the time-dependent receiver operating characteristic (AUROC) curves for 1, 3, and 5 years were 0.784, 0.801, and 0.806 in the model cohort, respectively (Fig. 4A). Survival curve analysis revealed that the mortality rate of patients in high-risk group was significantly higher than that in low-risk group in the model cohort (Fig. 4B). The calibration correction curves suggested that the predictive model demonstrated good consistency between the predicted and actual mortality rates in the model cohort (Fig. 5).

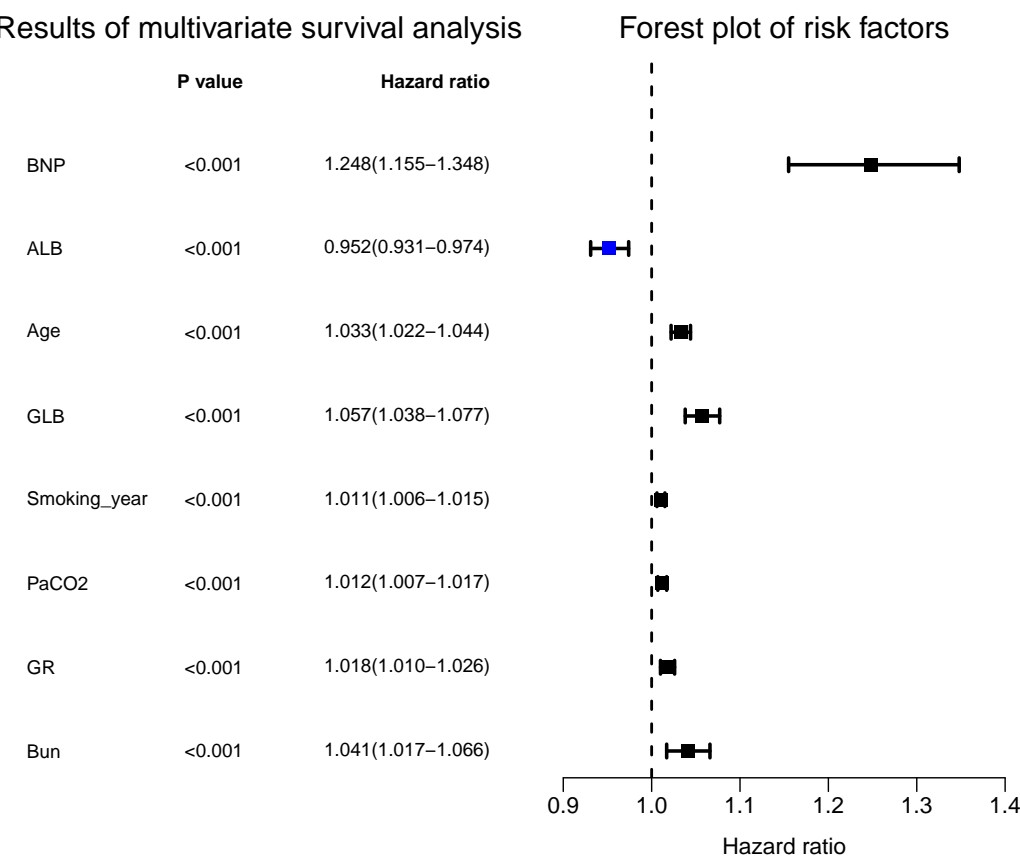

**Figure 2** Independent risk factors for the prognosis in multivariate Cox proportional hazards regression analysis.

## Performance in the validation cohort

For the validation cohort, the AUROC curves for 1, 3, and 5 years were 0.765, 0.779, and 0.798, respectively (Fig. 6A). Survival curve analysis revealed that the mortality rate of patients in the high-risk group was significantly higher than that in low-risk group in the validation cohort (Fig. 6B). The calibration correction curves suggested that the predictive model demonstrated good consistency between the predicted mortality rate and the actual mortality rate in the validation cohort (Fig. 7).

## Online predictive tool

To help clinicians and COPD patients in using the predictive model in predicting the mortality risk curve of individual COPD patient, an online predictive tool, the "Smart Survival Predictive System for COPD patients", was developed (https://zhangzhiqiao15. shinyapps.io/Smart_survival_predictive_system_for_COPD/). The user can freely choose from among eight values on the interactive webpage, and then click the "prediction" button to obtain the individual mortality predictive curve for an individual COPD patient. A representative mortality risk predictive curve generated by the Smart Survival Predictive

**Table 2    Results of multivariate survival analysis of model cohort and validation cohort.**

| Parameters | Coefficient | HR | Lower 95% CI | Higher 95% CI | P value |
|---|---|---|---|---|---|
| Model cohort | | | | | |
| BNP | 0.221 | 1.248 | 1.155 | 1.348 | <0.001 |
| ALB | −0.049 | 0.952 | 0.931 | 0.974 | <0.001 |
| Age | 0.033 | 1.033 | 1.022 | 1.044 | <0.001 |
| GLB | 0.056 | 1.057 | 1.038 | 1.077 | <0.001 |
| Smoking_year | 0.010 | 1.011 | 1.006 | 1.015 | <0.001 |
| PaCO2 | 0.012 | 1.012 | 1.007 | 1.017 | <0.001 |
| GR | 0.018 | 1.018 | 1.010 | 1.026 | <0.001 |
| Bun | 0.040 | 1.041 | 1.017 | 1.066 | 0.001 |
| Validation cohort | | | | | |
| BNP | 0.194 | 1.214 | 1.128 | 1.308 | <0.001 |
| ALB | −0.081 | 0.922 | 0.904 | 0.941 | <0.001 |
| Age | 0.030 | 1.030 | 1.019 | 1.041 | <0.001 |
| GLB | 0.049 | 1.050 | 1.031 | 1.068 | <0.001 |
| Smoking_year | 0.008 | 1.008 | 1.004 | 1.012 | <0.001 |
| PaCO2 | 0.015 | 1.015 | 1.009 | 1.020 | <0.001 |
| GR | 0.016 | 1.016 | 1.008 | 1.024 | <0.001 |
| Bun | 0.029 | 1.029 | 1.006 | 1.054 | 0.015 |

**Figure 3    Mortality predictive nomogram chart.**

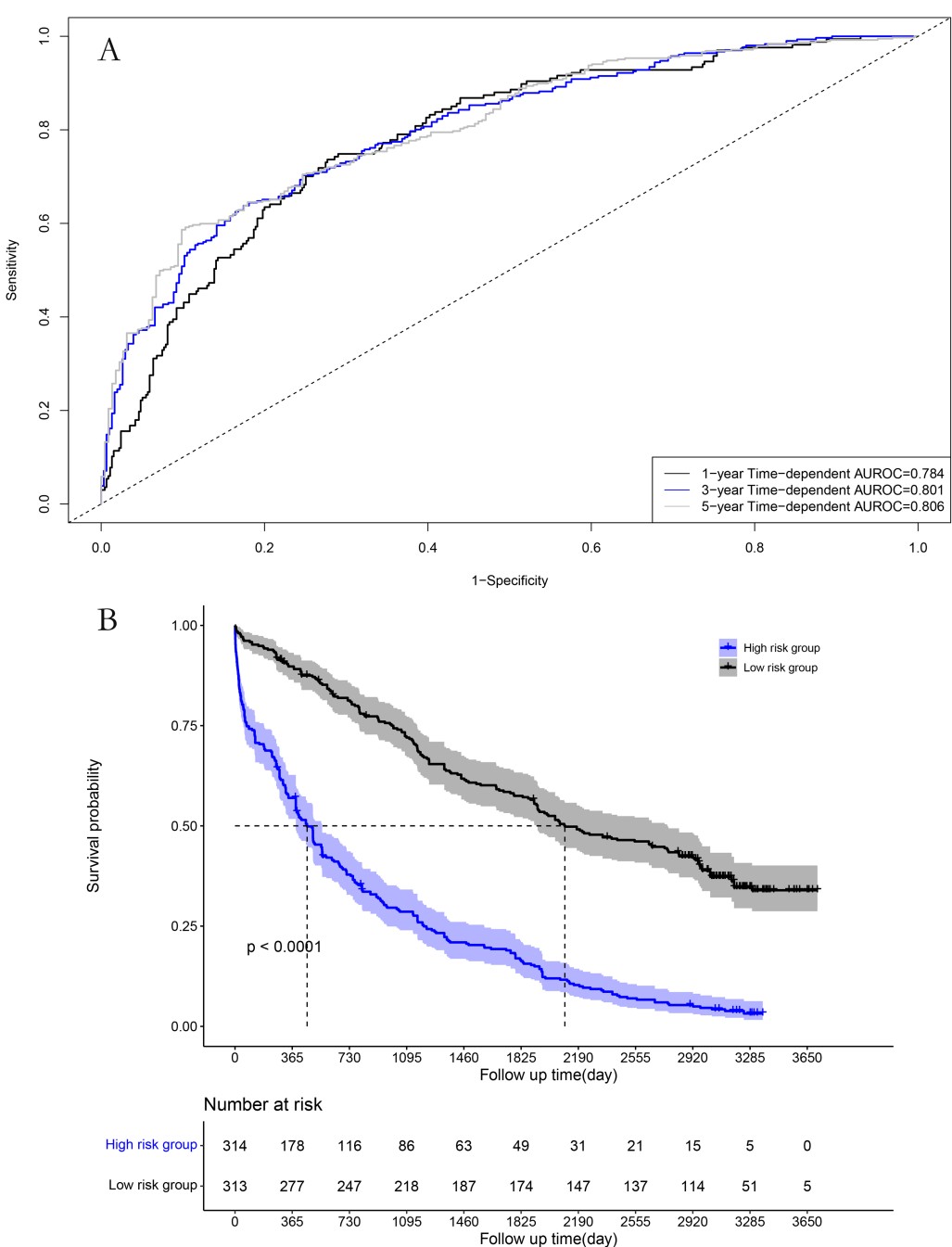

**Figure 4  Performance of prognostic model in the model group: (A) Time-dependent receiver operating characteristic curves in the model group; (B) survival curves in the model group.**

System for an individual COPD patient is shown in Fig. 8. In addition, the Smart Survival Predictive System for COPD patients can also provide the predicted mortality rate and 95% CI at a specific time.

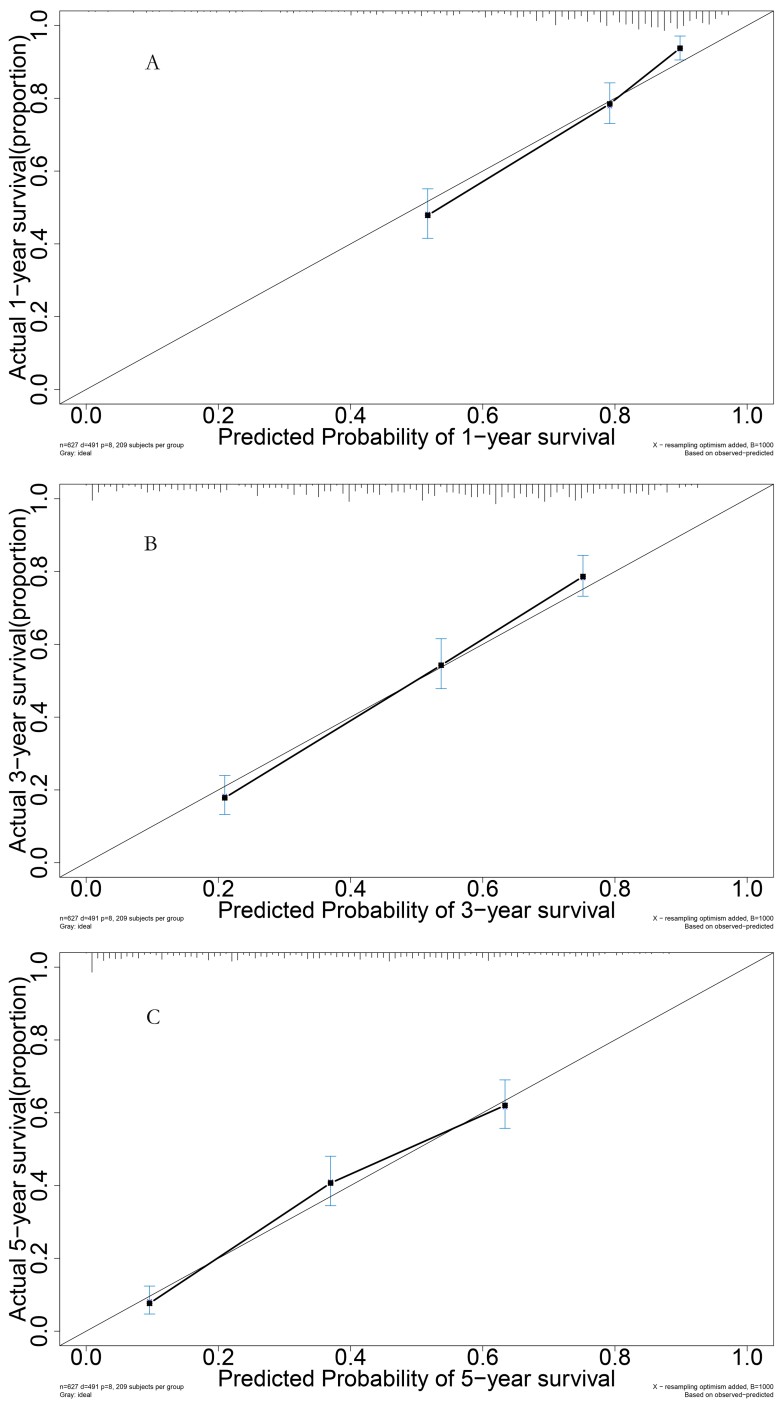

**Figure 5** Performance of prognostic model: (A) Calibration curve for 1-year in the model group; (B) calibration curve for 3-year in the model group; (C) calibration curve for 5-year in the model group.

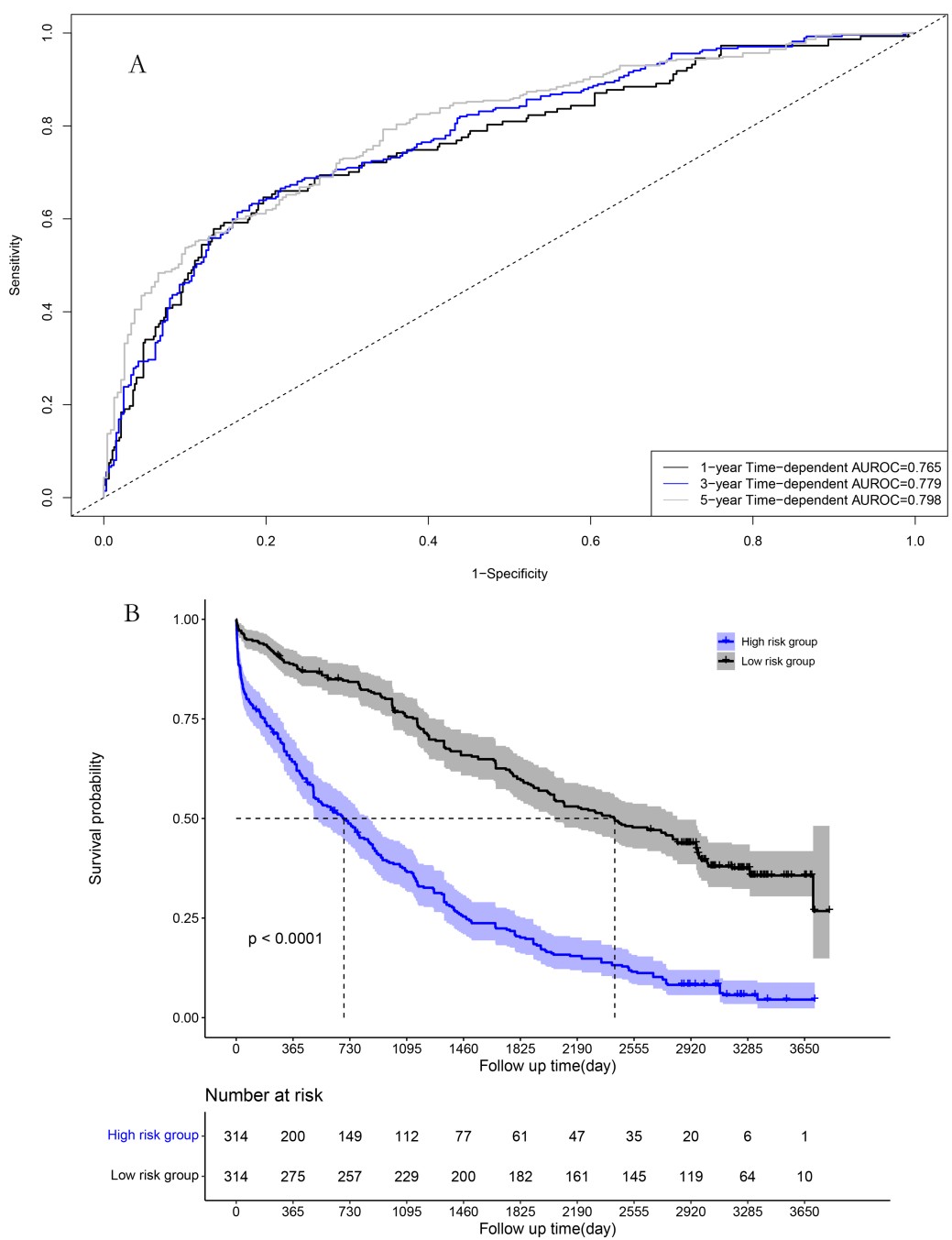

**Figure 6** Performance of prognostic model in validation group: (A) Time-dependent receiver operating characteristic curves in the validation group; (B) survival curves in the validation group.

## DISCUSSION

The current study identified eight independent risk factors for the prognosis of COPD patients according to the random survival forest model. Based on these eight risk factors, we constructed a prognostic model to predict individual mortality risk curves for subject-level

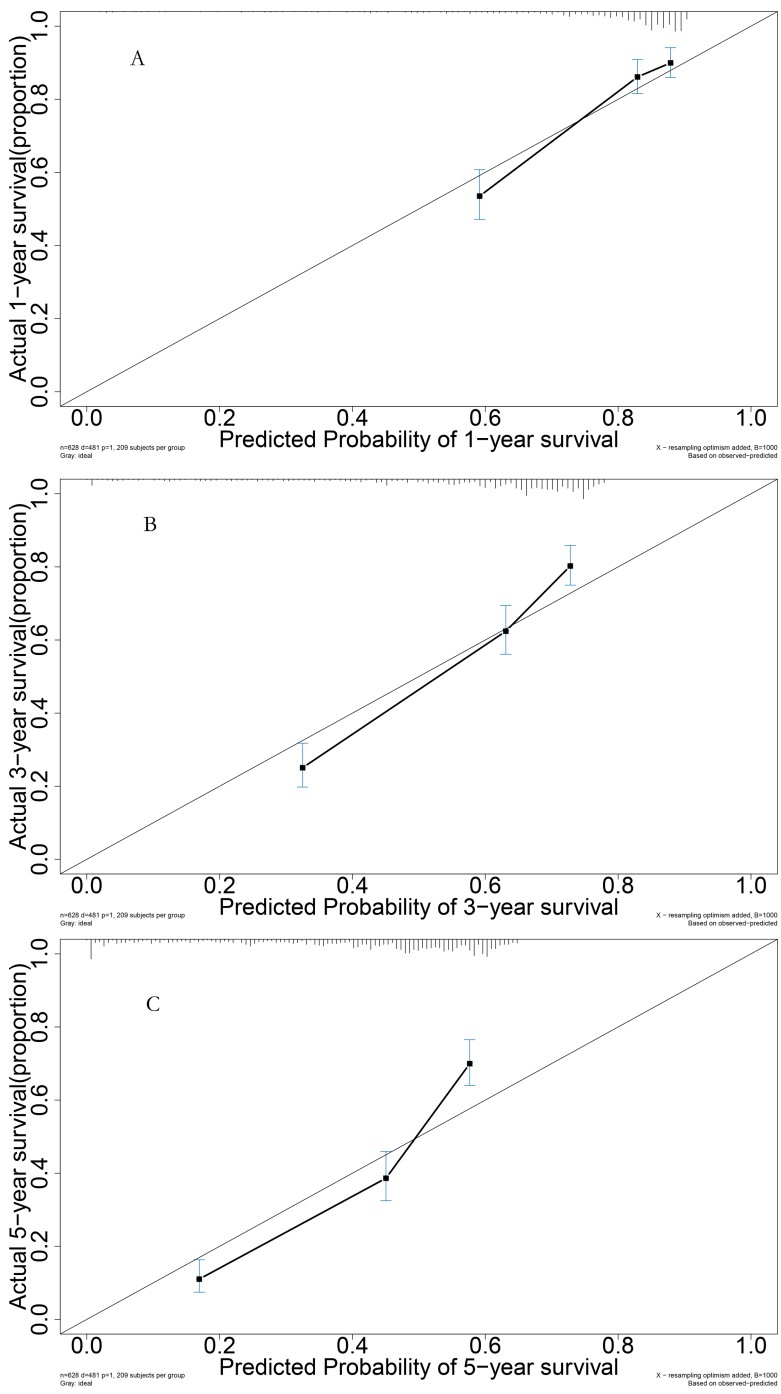

**Figure 7** Performance of prognostic model: (A) Calibration curve for 1-year in the validation group; (B) calibration curve for 3-year in the validation group; (C) calibration curve for 5-year in the valida-tion group.

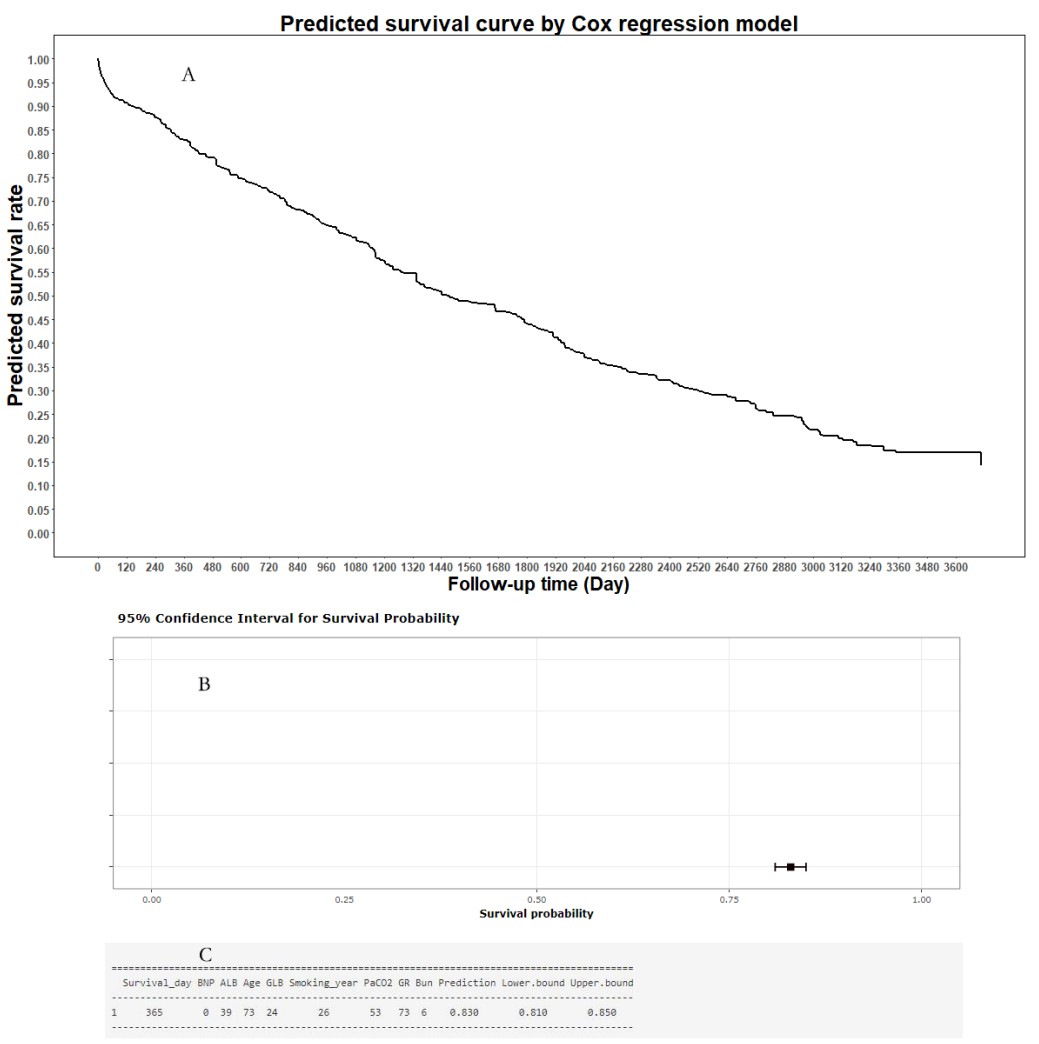

**Figure 8** Individual mortality risk prediction generated by Smart Survival Predictive System: (A) Individual mortality risk predictive curve; (B) distribution chart of individual mortality rate and 95% confidence interval at a special time-point; (C) individual mortality rate and 95% confidence interval at a special time-point.

COPD patients, and further developed a convenient web predictive tool. The AUROC curve and calibration correction curves suggested that the prognostic model demonstrated good predictive and discriminative ability in predicting the prognosis of COPD patients.

High BUN level at admission was an independent risk factor for death among COPD patients (*Chen et al., 2021*). BUN was related to the severity of disease in patients with COPD and could be used to assess the risk of prognosis (*Shorr et al., 2011*). $Pa\,CO_2$ was an independent risk factor for death of hospitalized COPD patients (*Hu et al., 2016*). The higher $Pa\,CO_2$ level in COPD patients suggested poorer prognosis than patients with lower $Pa\,CO_2$ level (*Wen et al., 2014*). Albumin and GLB could be used to evaluate the severity of disease and predict the risk for death in elderly patients with COPD (*Qin et al., 2018*).

Low albumin level was associated with poor 10-year survival in COPD patients (*Tang et al., 2021*). There was an independent correlation between albumin and the severity of illness in patients with COPD (*Li et al., 2021*). GLB level was associated with the severity of COPD patients and could be used to identify high-risk patients (*Li et al., 2021*). NT-proBNP may be a prognostic factor for poor prognosis of COPD patients (*Sánchez-Marteles et al., 2009*). BNP was an independent risk factor for secondary pulmonary hypertension in COPD patients, providing a valuable clue for the close relationship between the clinically poor prognosis of COPD patients and BNP (*Yang et al., 2019*). Age has been shown to be an independent risk factor and has been used to construct a prognostic model for COPD patients (*Puhan et al., 2012*). Age was a statistically significant independent risk factor for death among patients with COPD (*Shorr et al., 2011*). Smoking years has been used to predict the risk for deterioration of COPD patients (*Bertens et al., 2013*). A close relationship between smoke exposure and poor prognosis in COPD patients has been reported (*Golpe et al., 2015*). The neutrophil ratio has been used to predict in-hospital mortality of COPD patients (*El-Gazzar et al., 2020*). The neutrophil ratio was an independent risk factor of exacerbation in patients with COPD (*Ye et al., 2019*). The results of these previous clinical studies provide strong support for the eight independent risk factors found in our study.

The current study had several strengths. First, it constructed an online tool to predict mortality risk curves for individual COPD patients, which could provide subject-level risk prediction for this patient population. Second, it constructed a predictive nomogram chart to predict individual mortality risk at the subject-level of COPD patients at 1, 3, and 5 years using a Cox proportional hazards regression model algorithm. Third, the present investigation was a novel clinical study aiming to provide individualized predictive mortality risk curves for COPD patients, which provides a valuable avenue for exploration of individualized prognostic predictive study of patients with COPD.

Nevertheless, the present study also had some limitations, the first of which was the relatively small sample size ($n = 1255$), which may—to a certain extent—have affected the stability and reliability of the research results. As such, a larger sample size is necessary for future research to strengthen the reliability of the conclusions drawn. Second, because most patients enrolled in the current study were hospitalized with severe COPD, many did not undergo pulmonary function examination during hospitalization due to serious illness. As such, the final research data did not include real-time $FEV_1$ data and several other parameters during hospitalization. Third, several important clinical indicators were not included in the current study ($FEV_1$, modified Medical Research Council dyspnea scale, and 6 min walk distance test); thus, it is difficult to compare our results with several previously reported prognostic models. It is necessary for future research to include these previous important clinical independent variables and perform a comprehensive comparison with previously reported prognostic models. Fourth, because all patients in the current study were from China, the clinical applicability of our predictive model for individuals with COPD is not necessarily generalizable to the same patient population (s) in other geographical regions. Future research cohorts from different regions will help clarify the clinical applicability of the current predictive model to populations across different regions.

In conclusion, the current study constructed a prognostic model for predicting individual mortality risk curves for COPD patients after discharge and provided a convenient online predictive tool. This prognostic predictive tool demonstrated good ability to discriminate between high- and low-risk patients, and can provide valuable prognostic information for clinical treatment decision making during hospitalization and health management after discharge.

**Abbreviations**

| | |
|---|---|
| **ROC** | receiver operating characteristic |
| **HR** | hazard ratio |
| **CI** | confidence interval |
| **GR** | granulocyte ratio |
| **BUN** | blood urea nitrogen |
| **ALB** | albumin |
| **GLB** | globulin |
| **BNP** | B-type natriuretic peptide |

### Funding

The current research is a project supported by the Foshan Science and Technology Bureau (2020001005121). The funders had no role in study design, data collection and analysis, decision to publish, or preparation of the manuscript.

### Grant Disclosures

The following grant information was disclosed by the authors:
Foshan Science and Technology Bureau: 2020001005121.

### Competing Interests

The authors declare there are no competing interests.

### Author Contributions

- Shubiao Lu conceived and designed the experiments, performed the experiments, analyzed the data, prepared figures and/or tables, authored or reviewed drafts of the article, and approved the final draft.
- Yuwen Zhou performed the experiments, analyzed the data, prepared figures and/or tables, authored or reviewed drafts of the article, and approved the final draft.
- Xuejuan Huang performed the experiments, analyzed the data, prepared figures and/or tables, authored or reviewed drafts of the article, and approved the final draft.
- Jinsong Lin conceived and designed the experiments, performed the experiments, analyzed the data, prepared figures and/or tables, authored or reviewed drafts of the article, and approved the final draft.
- Yingyu Wu performed the experiments, analyzed the data, prepared figures and/or tables, authored or reviewed drafts of the article, and approved the final draft.

- Zhiqiao Zhang conceived and designed the experiments, performed the experiments, analyzed the data, prepared figures and/or tables, authored or reviewed drafts of the article, and approved the final draft.

## Human Ethics

The following information was supplied relating to ethical approvals (i.e., approving body and any reference numbers):

This study was reviewed and approved by the Ethics Committee of The Affiliated Chencun Hospital of Shunde Hospital, Southern Medical University (ID: 202202001). Due to the retrospective nature of the study and the use of anonymized data, requirements for informed consent were waived by the Ethics Committee of The Affiliated Chencun Hospital of Shunde Hospital, Southern Medical University (ID: 202202001).

## Ethics

The following information was supplied relating to ethical approvals (i.e., approving body and any reference numbers):

This article has been reviewed and approved by the Ethics Committee of The Affiliated Chencun Hospital of Shunde Hospital, Southern Medical University (ID:202202001). The informed consents were waved by the Ethics Committee of The Affiliated Chencun Hospital of Shunde Hospital, Southern Medical University because the current study is a retrospective cohort study and all individual information was anonymous from study datasets (ID: 202202001). The current study was conducted in accordance with the Declaration of Helsinki, relevant guidelines, and local regulations.

## Data Availability

The raw data is available in the Supplemental Files.

## Supplemental Information

Supplemental information for this article can be found online at http://dx.doi.org/10.7717/peerj.14457#supplemental-information.

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
