# Peer review of "Prediction of individual mortality risk among patients with chronic obstructive pulmonary disease: a convenient, online, individualized, predictive mortality risk tool based on a retrospective cohort study"

_PeerJ, doi:10.7717/peerj.14457_

## Round 0.1 · original submission · Major Revisions

The study aimed to construct a prognostic model for predicting the individual mortality risk of chronic obstructive pulmonary disease (COPD) patients, and provide an online web predictive tool so that COPD patients can more easily predict the individual mortality risk. This study may be interesting for researchers who are mainly interested in pulmonary disease. The manuscript has been reviewed and requires several modifications before making a decision. The comments of the reviewers are included at the bottom of this letter. Reviewers indicated that the methods section should be improved. They also indicated that your manuscript needs English editing. I agree with the evaluation, and I would request the manuscript be revised accordingly.

Reviewer 1 ·

Basic reporting

1. Line 153-156. It’s not clear what the authors mean “Cox model does not need to estimate the survival distribution type of original data, …”. The language is ambiguous. Please clarify.

2. Line 162-163. Continuous variables were compared between two groups by t-test or non-parametric method. Which non-parametric methods was used? Please specify.

3. Line 163-165. Chi-squared test or Fisher’s exact test was used to compare categorical variables between two groups. When was Chi-squared test used? And when for Fisher’s exact test? It needs to be clear. Please clarify.

4. Line 178. Random survival forest method was used, but it’s not mentioned in the Methods section. Please add details. In addition, random survival forest can also be used to construct a prognostic predictive model. Why did the authors choose the Cox model after using random forest screening risk factors? Did the authors compare the predictive performance of random survival forest and the Cox model?

5. For the risk factors included in the model, did the authors consider any potential non-linear or interaction effect?

6. Is the study cohort representative to the general population in China or any specific regions? I appreciate the authors’ development of the web tool, but it would be helpful for readers or users to understand if they can apply the tool to their patient population. If it cannot be applied to the general population, it’s important to indicate this limitation in the Discussion section.

7. Line 275-277. The authors claimed that this study is the first clinical study in the world to provide an individualized mortality risk predictive curve for COPD patients. To my understanding, the predictive curve is a byproduct of the Cox prediction model. It should be relatively straightforward to generate the predictive curve once the Cox model is built. Thus, I suggest the authors to reconsider if this claim is appropriate.

Experimental design

See comments above

Validity of the findings

See comments above

Reviewer 2 ·

Basic reporting

Lu et al. developed an online interactive tool for predicting the individual-level mortality risk curve of COPD patients after discharge. This tool applies a prognostic model based on Cox proportional hazards regression. Overall, the manuscript is clear, and the information is complete. The author provides details of the model development.

Here are some minor problems regarding the language, and I list some common errors that could be corrected in the manuscript.

1. Line 26, poor prognosis -> a poor prognosis;
2. Line 26, “at present ….” -> “Currently, no clinical study provides an individual level mortality risk curve for COPD patients.”;
3. Line 27, individual level -> individual-level;
4. Line 29, on-line -> online;
5. Line 32, 1255->1,255;
6. Line 32 – 33, “There were … in the current study” need to be rewritten for clarity;
7. Line 44 – 45, “A prognostic … factors.” Can be re-written as “A prognostic model for predicting death risk was constructed using these eight risk factors.”;
8. Line 48, on-line -> online;
9. Line 48, each word in “Survival curve prediction system” needs to be capitalized;
10. Line 192, why is it “+-”?
11. Line 215, has -> had

Experimental design

The details of the method development are provided in this manuscript. The data is described in detail. Can you add the information regarding the assumption check of data from the model cohort and the validation cohorts?

Validity of the findings

The manuscript has met all the requirements.

Additional comments

None

---

## Round 0.2 · accepted · Accept

Thank you very much for the submission of a revised version of your paper. I have gone through the revised, track-changes manuscript and rebuttal letter, and see that the authors addressed the reviewers' concerns and substantially improved the content of the manuscript. So, based on my own assessment as an academic editor, the manuscript may be now accepted for publication.